# Renewed: Protocol for a randomised controlled trial of a digital intervention to support quality of life in cancer survivors

Adele Krusche,[1] Katherine Bradbury,[1] Teresa Corbett,[2] Jane Barnett,[3] Beth Stuart,[3] Guiqing Lily Yao,[4] Roger Bacon,[5] Dankmar Böhning,[6] Tara Cheetham-Blake,[2] Diana Eccles,[7] Claire Foster,[8] Adam William Alfred Geraghty,[3] Geraldine Leydon,[3] Andre Müller,[9] Richard D Neal,[10] Richard Osborne,[11] Shanaya Rathod,[12] Alison Richardson,[2] Geoffrey Sharman,[5] Kevin Summers,[5] Eila Watson,[13] Laura Wilde,[14] Clare Wilkinson,[15] Lucy Yardley,[1,16] Paul Little[3]

For numbered affiliations see end of article.

**Correspondence to**
Dr Adele Krusche;
a.s.krusche@soton.ac.uk

## ABSTRACT

**Introduction** Low quality of life is common in cancer survivors. Increasing physical activity, improving diet, supporting psychological well-being and weight loss can improve quality of life in several cancers and may limit relapse. The aim of the randomised controlled trial outlined in this protocol is to examine whether a digital intervention (Renewed), with or without human support, can improve quality of life in cancer survivors. Renewed provides support for increasing physical activity, managing difficult emotions, eating a healthier diet and weight management.

**Methods and analysis** A randomised controlled trial is being conducted comparing usual care, access to Renewed or access to Renewed with brief human support. Cancer survivors who have had colorectal, breast or prostate cancer will be identified and invited through general practice searches and mail-outs. Participants are asked to complete baseline measures immediately after screening and will then be randomised to a study group; this is all completed on the Renewed website. The primary outcome is quality of life measured by the European Organization for Research and Treatment of Cancer QLQ-c30. Secondary outcomes include anxiety and depression, fear of cancer recurrence, general well-being, enablement and items relating to costs for a health economics analysis. Process measures include perceptions of human support, intervention usage and satisfaction, and adherence to behavioural changes. Qualitative process evaluations will be conducted with patients and healthcare staff providing support.

**Ethics and dissemination** The trial has been approved by the NHS Research Ethics Committee (Reference 18/NW/0013). The results of this trial will be published in peer-reviewed journals and through conference presentations.

**Trial registration number** ISRCTN96374224; Pre-results.

## Strengths and limitations of this study

► The intervention is designed to be easy to implement at scale.
► The intervention was developed using evidence-, theory- and person-based approaches.
► This trial will only be able to explore effects in patients with breast, colorectal and prostate cancer. However, this will enable exploration of whether the intervention might be a viable form of support for survivors of other forms of cancer.

## INTRODUCTION

Population-based studies indicate that quality of life (QoL) is poor in as many as one-third of cancer survivors post-treatment,[1] with levels equivalent to those in people living with major chronic diseases.[2 3] Those who struggle with aspects of QoL after treatment experience particular problems with psychological distress and fatigue[2] and social, physical and financial consequences,[4 5] which can persist for 10 years or more post-treatment.[5] This is a significant concern given the rising prevalence of secondary cancers due to heightened life expectancy and common environmental and lifestyle factors associated with increased risk, such as unhealthy diet.[6–8] Up to 75% of survivors of cancer also then struggle with comorbid issues such as arthritis, high blood pressure and anxiety and depression, which contribute to poorer quality of life.[9] Evidence suggests that early intervention could alleviate some longer term problems and reduce burden on the health service[10] and that specifically addressing physical activity, psychological well-being, diet

and weight loss is likely to improve QoL in survivors of a range of cancers and may limit relapse.[11–14]

### Physical activity

The positive effects of physical activity in cancer survivors are well documented. Meta-analyses indicate that physical activity interventions reduce fatigue in cancer survivors.[15–18] Overall, positive trends and impact of physical activity interventions exist for physiological, psychosocial and functional outcomes.[19 20] As a result, being sufficiently active is one of the cornerstones of a long-term self-care strategy for cancer survivors.[21]

### Psychological Well-being

The psychological well-being of cancer survivors can be enhanced through a range of approaches. Cognitive behavioural therapy (CBT) techniques aimed at survivors have been found to significantly reduce psychological distress, including depression and anxiety.[22] Mindfulness-based interventions place emphasis on the acceptance and awareness of the present through the use of mindfulness techniques, including meditation exercises.[23] Review evidence indicates mindfulness-based techniques can reduce stress and anxiety in breast cancer survivors[24] and improve QoL in cancer survivors, often by improving fear of cancer recurrence, stress, anxiety and depression.[25] Mindfulness-based interventions also effectively reduce cancer-related fatigue.[26 27]

### Diet

Improving diet can positively impact the health and well-being of cancer survivors. Lowering fat consumption reduces cancer relapse[28] and data from the Healthy Survey for England indicates an association between increased fruit and vegetable intake and reduced cancer mortality. Systematic review evidence also indicates that healthy dietary changes improve health-related QoL in cancer survivors.[29]

### Weight management

Being overweight or obese is common among cancer survivors[15] and associated with negative health and well-being consequences, including risk of cancer recurrence.[30 31] Review evidence suggests that weight loss is feasible and safe for cancer survivors.[32] In addition, breast cancer-related biomarkers reduced by 30%–40% in breast cancer survivors who lost 5% or more of their body weight, indicating a lower risk of cancer recurrence.[30] Weight loss also has a positive effect on QoL in cancer survivors.[32]

### Using the internet

The current study will evaluate a digital intervention created specifically to improve QoL in cancer survivors. A digital intervention is likely to be an efficient and cost-effective mode of delivery of an intervention to support cancer survivors in improving their QoL. The internet is now used extensively and successfully by older adults for disease self-management[33] and a recent review found that internet-based interventions

can be as effective as face-to-face therapy, becoming a viable cost-effective alternative.[34]

## AIMS AND HYPOTHESES

The aim of the Renewed trial is to evaluate the effectiveness and cost-effectiveness of two trial arms using an intervention designed to improve QoL in breast, colon and prostate cancer survivors: the first arm will be given access to a digital intervention (Renewed) and the second will be given access to Renewed accompanied by brief human support.

The research aims to answer the following questions:

### Primary research question

1. Does the Renewed intervention, with or without human support, result in a difference in QoL (as measured by the European Organization for Research and Treatment of Cancer (EORTC) QLQ-c30;[35]) at 6 month follow-up compared with treatment as usual?

### Secondary research questions

1. Does the Renewed intervention, with or without human support, result in a difference in QoL at 12 month follow-up compared with treatment as usual?
2. Is the Renewed intervention, with or without human support, more cost-effective than usual care in improving QoL in cancer survivors?
3. Does the Renewed intervention, with or without human support, result in a difference in psychological and overall well-being (including low mood and fear of cancer recurrence) at 6 month and 12 month follow-up?
4. Does the Renewed intervention, with or without human support, help to reduce the risk of secondary cancers (recurrence, metastatic cancer or another form of cancer)?

### Qualitative process analysis research questions

1. Is the Renewed intervention, with or without human support, acceptable to patients and healthcare practitioners?
2. Is the Renewed support feasible for healthcare practitioners to implement in practice?

## METHODS

### Study design

The Renewed study is a pragmatic, randomised controlled trial comparing usual care, the Renewed intervention and Renewed accompanied by brief human support. The long-term aim of the intervention is to potentially help cancer survivors with other forms of cancer. At this stage, we have chosen three contrasting common survivor groups, who have no metastatic disease, to deal with the likely varying issues in needs and preferences across gender and age spectrums: breast cancer survivors (younger and older women); prostate cancer survivors and those on active surveillance/watchful waiting for prostate cancer (predominantly older men); and colorectal cancer (a

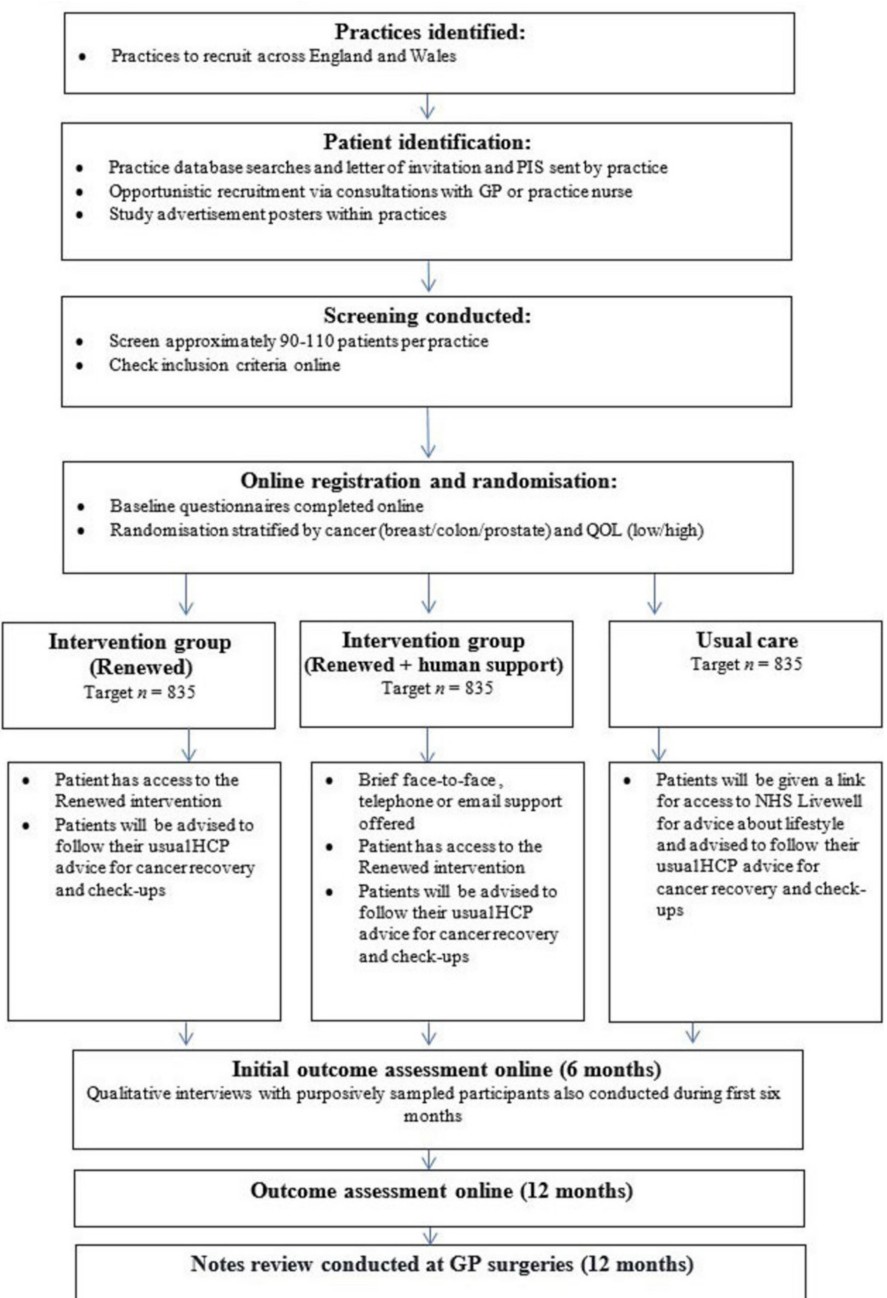

**Figure 1** Randomised controlled trial and pilot study procedure flow chart. GP, general practitioner; NHS, National Health Service; QoL, quality of life.

range of age and gender). By working with these groups, we will explore to what extent the intervention is likely to be generalisable to other survivor groups. See figure 1 for an outline of the flow of the study.

### Patient and public involvement

When first designing the study, the research team received input from six patient representatives who had experienced breast, prostate or colorectal cancer, in addition to members of Macmillan Cancer Support, Prostate Cancer UK, Breast Cancer Care, Bowel Cancer UK, Cancer Research UK and the National Cancer Research Institute. All parties had input into the original grant proposal and

played a part in developing and agreeing the research questions, study procedures (eg, inclusion criteria) and choice of primary outcome.

After securing funding, an additional two patient representatives were recruited to replace two patient representatives who were unable to continue their involvement due to ill health. Our panel of six patient representatives and representatives from Prostate Cancer UK, Breast Cancer Care and Bowel Cancer UK are part of the study management team and attend regular meetings to contribute to the development of the Renewed intervention and the study protocol and procedures for the randomised controlled trial. They

assisted in refining the study design, for example, making suggestions for the frequency and mode of support delivery for patients randomised to the support arm, which were implemented. The patient representatives also took part in qualitative interviews and provided detailed comments on draft intervention materials (in addition to feedback obtained by qualitative interviews with 33 research participants recruited from primary care (see Bradbury et al, in prep). This feedback was used to inform modifications to the intervention to maximise engagement and minimise any burden posed by the intervention.

At the end of data collection, patient and charity representatives will be involved in the discussion and interpretation of the trial and process evaluation data and in planning dissemination with stakeholders.

### Recruitment

Participants (n=2500) will be recruited across England and Wales from sites in Southampton, Oxford and Bangor, via general practitioner (GP) surgery mail-outs. Practices will be informed about the study via clinical research networks and practices who opt-in can proceed to participate. Practice staff will search GP databases for eligible patients (see table 1 inclusion/exclusion criteria).

Lists of potentially eligible patients will be screened by GPs prior to mail-out. Anonymous data (age, cancer type, postcode and diagnosis date) will be collected via GP record searches on all invited patients to allow exploration of the generalisability of our sample.

Patients will be mailed a letter of invitation, an information sheet, instructions on how to start the study and a reply slip collecting reasons for non-participation. Opportunistic recruitment by clinicians in participating surgeries will also be used to recruit patients who fit the inclusion criteria. Posters advertising the study may also be presented in waiting areas. The expected dates of recruitment run from October 2017 until July 2019.

### Eligibility criteria

See table 1. We will be using a cut-off of 85 on the primary outcome measure for QoL (where 100 is the best possible score), the EORTC QLQ-c30[35] so that those scoring above 85 (who have high quality of life) will be excluded. This score was calculated using existing data examining the change in QoL scores for cancer survivors, where the lowest scoring two-thirds scored 87.1 or lower, with a median of 80.9[36] (higher scores denoting higher QoL). Participants will be excluded if they have had sarcoma or lymphoma of the breast as these have a different course, prognosis and treatment, as agreed by the clinical members of the research team. Those living in the same household as another study participant will also be excluded to avoid potential contamination across study groups.

### Sample size and power calculation

The primary outcome is QoL as measured by the total EORTC score at 6 months. The study is powered to detect a standardised effect size of 0.3, consistent with

**Table 1** Inclusion/exclusion criteria

| Criteria | Screened by general practitioner | Screened online |
|---|---|---|
| **Inclusions** | | |
| Aged at least 18 years | x | x |
| Had a diagnosis of colorectal, breast or prostate cancer | x | x |
| Finished primary cancer treatment within prior ten years/ diagnosis in prior ten years, unless on active surveillance for prostate cancer | x | |
| Have internet access | | x |
| Impaired quality of life (≤85 on the EORTC) | | x |
| **Exclusions** | | |
| Have had more than one type of cancer in the preceding 5 years | x | x |
| Has metastatic cancer | x | |
| Has sarcoma or lymphoma of the breast | x | |
| Receiving cancer treatment or had recent treatment (in last month) | x | x |
| Expecting to start cancer treatment during the study period | | x |
| Severe mental health problems and/or major uncontrolled depression/schizophrenia or dementia | x | |
| Lives in the same household as another participant | x | |

Primary treatments can include bone marrow and stem cell transplants, breast-conserving surgery, chemotherapy, colostomy/partial colostomy, immunotherapy, lumpectomy, mastectomy, orchidectomy, radical prostatectomy, radiotherapy, targeted therapy, trans-urethral resection.
EORTC, European Organization for Research and Treatment of Cancer.

our previous internet-based behavioural interventions.[37] The key pairwise comparison is each of the two intervention groups versus controls. To detect a difference of 0.3 standard mean difference in any pairwise comparison between intervention and control for 80% power and alpha=0.05 requires 176 intervention participants in each intervention group and 176 controls, giving a total of 528 participants per cancer type. In order to explore the difference in the key subgroups (breast, colon and prostate cancers), this requires that we include 528 participants in each of the three clinical groups. This gives a total of 1584 participants, or 1980 allowing for 20% loss to follow-up. Even though this is an individually randomised design cluster effects are possible: if we assume eight patients per intervention group per practice then to

allow for clustering at a practice level and assuming an intraclass correlation coefficient (ICC) of 0.03 then the inflation factor is 1.21 (1+ (8–1*0.03)), which will require 2396 participants. Allowing for some leeway in our assumptions, we will aim to recruit approximately 835 patients per cancer type (total n=2500).

## Randomisation and blinding

After online screening, patients will complete online consent and baseline measures. LifeGuide software will then randomise each participant to one of the three study groups using computer-generated random numbers. Once randomised, participants will be informed of their allocation and, if in one of the intervention arms, given access to Renewed. Randomisation will be stratified by cancer type: breast/prostate/colon and by EORTC score (high/low QoL; 64 or less/65–85 taking 65 as the lower 25% CI from previous study data[36]). Blinding of randomisation is ensured by the use of automated computer software. Blinding of patients to the intervention is not possible. Participants will be informed online as to which group they have been allocated to immediately and will also be sent notification via email. Practice staff involved in the support arm of the trial will be notified by email when one of their patients is randomised to the Renewed + Support arm of the trial so that they can provide support.

## Renewed intervention

Participants randomised to the intervention groups will have access to their usual medical care and have access to the Renewed intervention.

The Renewed intervention was created using LifeGuide software (www.lifeguideonline.org).[38 39] During the development of Renewed a synthesis of the growing evidence base was conducted that relates to digital interventions directed at improving QoL in cancer survivors.[40] Consistent with best practice in digital behaviour change intervention development,[41 42] iterative qualitative research was used to elicit user (patient and health professional) views and experiences of the intervention modules throughout development, including barriers and facilitators to engagement with altering diet and exercise.[43] The content of Renewed has been refined based on these findings.

See figure 2 for a detailed diagram showing the flow of participants through the intervention, incorporating Template for Intervention Description and Replication (TiDIER) guidelines.[44] There are four interventions within Renewed that participants can choose to view: Getting Active for increasing physical activity, Healthy Paths for stress reduction, Eat for Health for diet improvement and POWeR+[45] for weight loss. In the introductory pages, Renewed provides participants with tailored suggestions of the aspects of the website which might be most helpful to them, based on their answers to the baseline QoL measure (EORTC). There is also advice and information about active surveillance/watchful waiting for prostate cancer for men being monitored. After the introduction, participants are presented with their homepage containing buttons to the separate interventions with some brief information about what to expect from each; POWeR+ is only presented to participants who have a BMI of over 25.

Participants are able to come back to the intervention and view which content they prefer as often as they choose. If a participant accesses one of these interventions they will be sent brief automated emails as reminders and for motivation with tips, interesting facts and brief information and advice. Content is tailored throughout Renewed, where appropriate, to present information most relevant to specific cancer types (eg, concerns about exercising with a colostomy bag) and the developers of the interventions worked together to ensure that conflicting advice is not given. Where appropriate, Renewed links to existing helpful websites rather than duplicating information: links to external resources are provided throughout the Renewed intervention, some of which are tailored by cancer type. For example, these include information about returning to work and connecting with other survivors of cancer (from the homepage) and links to websites providing further emotional support on Healthy Paths and websites about group exercise on Getting Active.

## Getting Active

Getting Active is designed to help participants increase their physical activity. It includes a quiz designed to increase motivation for physical activity by highlighting the benefits of increasing activity. Getting Active encourages gently increasing physical activity, taking a graded approach. Content addresses common barriers and concerns (eg, feeling tired, co-morbidities). There are various activity options such as exercising at home and walking. Users can set and review physical activity goals and receive tailored feedback on their progress.

## Healthy Paths and Healthy Mind

Healthy Paths aims to reduce stress and improve psychological distress through the use of mindfulness-based and CBT techniques.[46] It was amended for the current study to include specific information for cancer survivors about fear of recurrence and feelings of loss following cancer. The role of the stressor is acknowledged and patients are encouraged to use a variety of emotional regulation strategies to develop effective coping skills. There are a number of mindfulness-based exercises as well as behavioural activation strategies[47] and cognitive exercises focusing on emotion regulation (positive thought logging and self-compassion exercises).

Healthy Mind is a 'lite' version of Healthy Paths in the form of an app, which includes similar techniques to Healthy Paths (ie, CBT elements and mindfulness-based practices), but is for more general stress reduction and does not address cancer-specific concerns.

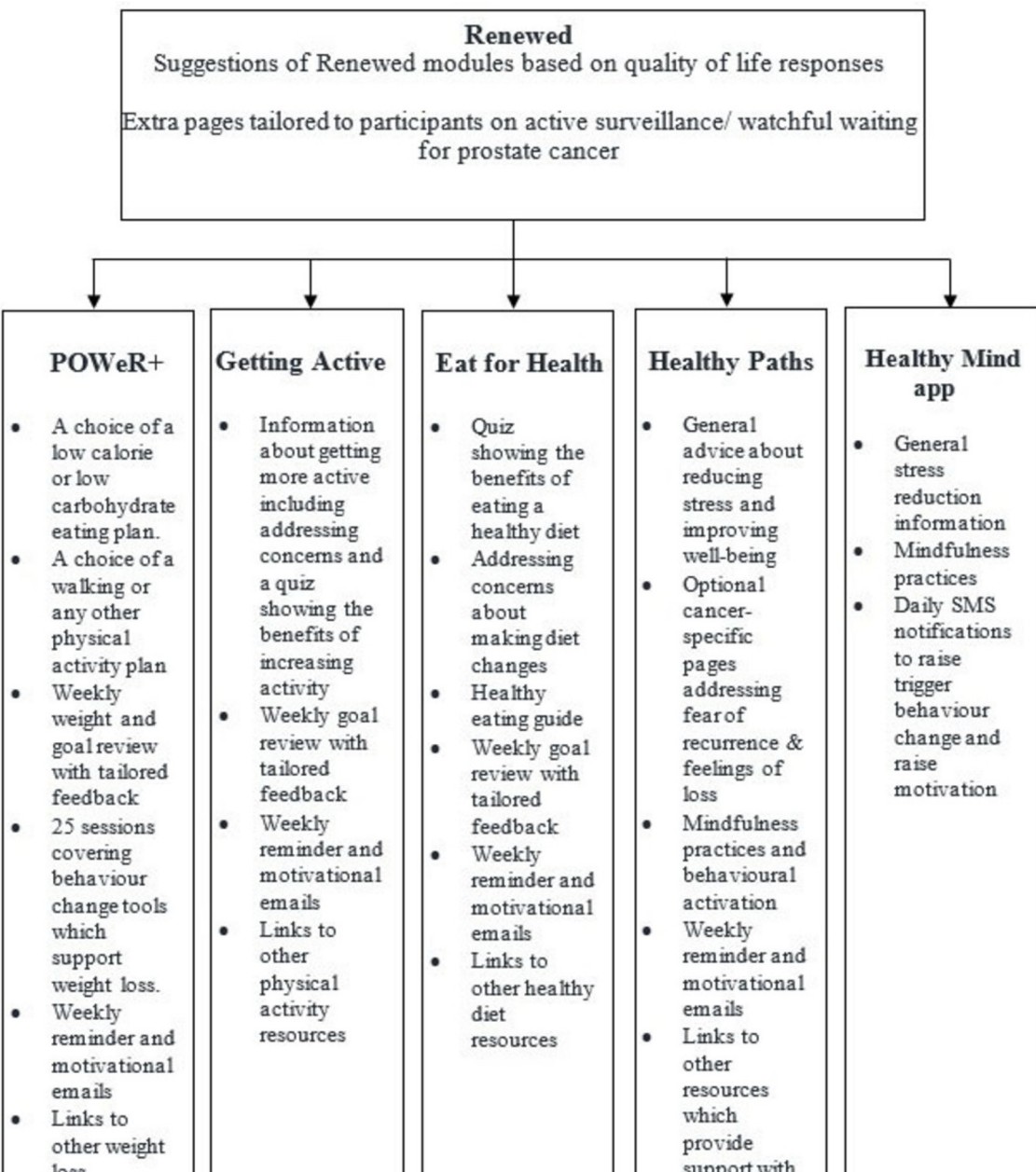

**Figure 2** Renewed intervention outline.

## POWeR+

POWeR+is an online weight management intervention, which has been shown to be effective and cost-effective, described in full elsewhere.[45] In brief: Participants can follow a low-calorie or low-carbohydrate eating plan and can choose between increasing walking and any other physical activities. Participants set themselves goals and review weight, goals and plans on a weekly basis, receiving tailored feedback on progress. Participants also have access to 25 sessions, which include topics such as coping with cravings, gaining social support and relapse prevention.

## Eat for Health

Eat for Health is based on a healthy diet (high in fruit and vegetables; reduced fat, sugar, alcohol, red and processed meats), which can reduce mortality rates in cancer survivors.[48] It differs from POWeR+ in that the focus is on eating a healthy diet rather than losing weight and is designed to enhance cancer survivor knowledge of healthy eating and increase motivation to make changes to eating habits. Eat for Health includes a short quiz about the benefits of healthy eating, an easy to follow eating plan using a traffic light system, patient stories modelling overcoming challenges, meal plans and healthy recipes. Participants are able to set, review and change their eating goals and receive personalised feedback on their progress.

## Renewed accompanied by brief human support

Participants in the Renewed group with accompanying brief human support will receive support from a nurse or healthcare assistant (at their surgery where possible, or else

a central support facilitator or research nurse employed by the study team where the practice are unable to provide their own support). 'Supporters' are able to view the Renewed intervention but do not need specific knowledge of Renewed content: Supporters will receive brief web-based training to learn how to provide support using the CARE approach (Congratulate, Ask, Reassure, Encourage[49 50]), which aims to help build patients' autonomous motivation for engaging with the digital intervention and offline behavioural changes (eg, getting more active).

Three 10 minute support sessions will be offered in person, by phone or email. The first will be offered 2 weeks after initially logging on to Renewed. Supporters are also asked to send encouraging, supportive emails at weeks 2 and 4 after randomisation and templates are supplied for this. Follow-up support will be offered at 4 and 8 weeks after baseline. Not all participants will want support and they can choose whether or not to contact their Supporter; Supporters will keep a record of the frequency and nature of support they provide to participants. Participants can contact their Supporter through the website. Number and type (email/phone/face-to-face) of support contacts, as well as length of appointments, will be recorded.

### Usual care
Participants allocated to usual care will receive a link to the National Health Service's (NHS) LiveWell website where they can access advice about living a healthier lifestyle. Other than this, participants in the usual care group will continue to use their existing medical support as they usually would. At the end of the study, participants will be asked what other interventions, if any, they have used while taking part in the study.

### Data collection
Participants will receive three automated notifications to complete follow-up measures at 6 and 12 months, one being the initial invitation to ask them to complete the online questionnaires and reminders at 1 week and 2 weeks later if they are not complete. Participants will receive a £10 gift voucher via email when they complete the 6 month follow-up questionnaires online. If, after 6 weeks, participants have not completed the 6 month follow-up questionnaires online, paper-based questionnaires will be sent with a £10 gift voucher for high-street stores (regardless of paper-based questionnaire completion). If paper-based questionnaires are not returned after 2 weeks of being sent, research staff (blind to group allocation) will phone participants to request limited responses over the phone (the EORTC); a maximum of two phone contacts per participant will be made. The procedure will be the same for 12 month follow-ups but with no further vouchers given. If at any stage the participant indicates that they would not be willing to complete any further measures, no further contact will be made regarding the follow-ups. At study completion, the usual care (wait list) group will be given access to the Renewed

website. Participants will be asked if they would like a copy of the study results, and if so whether they would prefer to receive these by email or post.

### Sub-study examining follow-up questionnaire design
There has been extensive research into how the design and presentation of questionnaires affects response rates and response quality (see[51] for an overview). Two versions of the 12 month outcome measures will be used, to test whether completion rates can be increased by enhanced presentation. Users will be randomised to either the control ('normal') outcome measures or the enhanced outcome measures. For both versions, the wording and scales will remain the same, but the presentation (eg, use of images, page layout and messages encouraging completion) will be altered for the enhanced version.

### Measures
Table 2 presents a list of the measures and when they will be presented to participants. Measures will be collected via the website.

**Table 2** Measures and times presented

| Measure | Baseline | Six months | 12 Months |
|---|---|---|---|
| **Primary outcome** | | | |
| EORTC QLQ-c30 | x | x | x |
| **Secondary outcomes** | | | |
| HADS | x | | x |
| FRRS | x | | x |
| EQ-5D-5L | x | x | x |
| PA | x | | |
| MYCAW | x | | x |
| Personal costs data for heath economics | | | x |
| **Process measures** | | | |
| TAQ | | x | |
| PEI | | | x |
| Website satisfaction and usage | | | x |
| PETS | | | x |
| Usage of other interventions during past year | | | x |

Demographics include gender, age, marital status, years of education, ethnicity, height and weight and are taken at baseline; EQ-5D-5L, Euro-Qol; EORTC QLQ-c30, European Organization for Research and Treatment of Cancer Quality of Life Questionnaire-c30 as primary outcome; FRRS, Fear of Relapse/Recurrence Scale; HADS, Hospital Anxiety and Depression Scale; MYCAW, Measure Yourself Concerns and Well-being; PA, Physical activity checker; PEI, Patient Enablement Instrument; Website satisfaction measure; PETS, Problematic Experiences of Therapy Scale; TAQ, Treatment Appraisal Questionnaire.

The primary outcome of QoL will be measured using the total score on the EORTC QLQ-c30.[35] The EORTC has 30 items and contains five functioning subscales: physical, emotional, social, role and cognitive and eight symptom subscales: pain, fatigue, nausea and vomiting, dyspnea, sleep problems, loss of appetite, constipation and diarrhoea, financial impact and overall quality of life. Each item has a four-point response scale from 'not at all' to 'very much' and scores are calculated using a linear conversion to create a score from 0 to 100. It is one of the most widely used measures for health-related QoL in cancer research.[52]

The Fear of Relapse/Recurrence Scale[53] has been modified to three items so that 'I will probably have a relapse in the next 5 years' and 'I am certain that I have been cured of cancer' are omitted as they do not apply to those on Active Surveillance. The Patient Enablement Instrument (PEI) was modified so that it asks about confidence and understanding relating to health as a result of using the website as opposed to receiving support from a doctor (PEI[54 55]). Data regarding website and/or app usage (eg, pages accessed, time spent on each page) will be downloaded using LifeGuide software. A medical record notes review will be conducted after 12 month follow-up to extract NHS and Personal Social Service use. This will enable collection of data pertaining to clinical status including cancer recurrence. Online questions will be used to collect patients' personal costs. The EQ-5D-5L[56 57] which measures medical QoL (QALY) will be collected at baseline, 6 months and 12 months. We will apply the UK tariff to translate the EQ-5D-5L to utility scores.

### Statistical analysis

SPSS, Stata and Excel software will be used to evaluate outcomes. All participant data will be analysed, including those who have withdrawn, unless participant/s specifically request that their data be removed from the data set. All participants will be analysed on an intention to treat basis, that is, as randomised with any missing data imputed using a chained equations multiple imputation model. Complete cases analysis will be undertaken as a sensitivity analysis.

Linear regression models will be used for the analysis of continuous variables, controlling for baseline values, stratification variables, practice as a cluster variable, and also controlling for potential confounding variables as appropriate, both for the overall trial sample and for each clinical subgroup (breast, prostate, colon). Results will be produced for the trial overall and within each cancer type. We will undertake pre-specified subgroup analyses, including exploring possible mediators and moderators in the context of the process analysis. These analyses will be set out in full in the Statistical Analysis Plan prior to the final follow-ups.

To examine how the design and presentation of questionnaires affects response rates and response quality, we will compare the initial completion rate for the control outcome measures versus the enhanced outcome measures using a factorial design where participants will be allocated to see the normal or enhanced questionnaires. A ratio of 1:2 (normal:enhanced) will be used when allocating participants to get a feel for how the enhanced questionnaire presentation might impact on study completion, given that unenhanced questions are already presented at baseline and 6 months.

### Health economics

Published unit costs (Personal Social Services Research Unit, British National Formulary and national reference costs) will be applied to itemised resource usage in calculating total cost per person. The health economic analysis will include cost effectiveness (£/unit of primary outcome) and cost utility (£/EQ-5D-5L QALY) analysis. The costs of the study will be from an NHS perspective with exploration of a quasi-social perspective including the time required by patients interacting with the intervention. The study will identify and quantify the resources required to deliver the intervention, and any impact on NHS service usage and personal costs. We will explore the potential change in NHS service use and change in the pattern of use of such services, such as whether the new intervention will reduce medication (eg, depression related drugs), GP consultations and community nurse and special service usage. The key resource usage of the intervention covers nursing support time, software. Personal costs will be collected on time off work (both patients/carers), cost of use of internet (personal time) and costs of over-the-counter medications. The economic analyses of both costs, QoL (EORTC and QALY), provided minimally worthwhile improvements are shown, will be summarised in terms of incremental cost effectiveness ratios and cost effectiveness acceptability curves.

### Qualitative process studies

There will be two qualitative process studies. One will examine participants' experiences of using the intervention and participating in the trial. Participants will be emailed the information sheet with an invitation to take part in this additional study. Eight to ten participants per cancer type (n=24–30) will be purposively sampled until saturation (by cancer type, age and gender). Participants will be invited to take part after they have taken part in the trial for at least 1–2 months. Additional consent will be sought for these studies. Interviews (face-to-face or by phone) will consist of open-ended questions to elicit detailed responses. Each interview will last approximately 60 min. These interviews will enable the research team to further assess the acceptability and helpfulness of the intervention and any barriers to engagement. Control participants will also be invited in the same way to participate in brief interviews to elicit qualitative feedback about the brief advice link they were given at baseline (the NHS LiveWell website) and the study.

The second qualitative process study will be conducted with Supporters, to explore their views of supporting patients in using Renewed online. This study will elicit perceptions of the initial training website, providing support to patients and the general study procedures.

After at least 1–2 months of participation, support providers, GPs and any other primary care staff significantly involved in trial procedures or intervention delivery (n=20, or until saturation) will be invited to take part in this additional study. Focus groups, interviews or phone interviews will be held, depending on availability or else electronic (written) feedback will be requested.

In both process studies interviews will be recorded and fully transcribed. Transcriptions will be anonymised. To ensure that we remain open to, and grounded in, interviewee perspectives we will carry out inductive thematic analysis of all textual data,[58] triangulated where appropriate with other trial data (eg, web usage), and with discussion among team members (including our PPI representatives) to elaborate our interpretations.[58]

## Handling of adverse events

It is very unlikely that there will be any adverse events during screening and questionnaire completion, given that screening online consists of answering questions; however, participants are repeatedly told that they can contact the research team who can signpost them to other support services as appropriate. Participants have the right to withdraw from the study at any time. If a participant withdraws having completed questionnaires, their data will be retained to evaluate potential differences and reasons for attrition, unless they ask us not to use their data. Should a participant be ineligible for the study for any reason, they are presented with a page of alternative support services.

Any serious adverse event (SAE) occurring to a research participant will be reported to the Ethics Committee where, in the opinion of the Chief Investigator, the event was related to administration of any of the research procedures, and was an unexpected occurrence. Non-serious AEs will not be collected. Pre-planned hospitalisation for example, for pre-existing conditions which have not worsened or elective procedures for a pre-existing condition will not be classed as an SAE. GP surgeries will inform the research team and/or the clinical trials unit of any SAEs within 24 hours of becoming aware of the event occurring. GP surgeries will be provided with a standard operating procedure and a form for SAE reporting. Reports of SAEs will be provided to the Committee within 15 days of the Chief Investigator becoming aware of the event. All SAEs will also be sent to the Trial Steering Committee.

## Dissemination

The results of this trial will be published in peer-reviewed journals and through conference presentations. Press releases will be utilised to disseminate the findings to the general public and the findings will also be sent to the GP surgeries who participate and to those participants who request them. Social media networks connected to the research group (eg, Twitter) will be used to disseminate published research. If proven effective, Renewed will be developed with service providers and charities to explore mechanisms through which it can be made available to patients.

**Author affiliations**
¹Department of Psychology, University of Southampton, Southampton, UK
²School of Health Sciences, University of Southampton, Southampton, UK
³Primary Care and Population Sciences Division, University of Southampton, Southampton, UK
⁴Biostatistics Research Group, University of Leicester, Leicester, UK
⁵Patient and Public Involvement team for the CLASP project
⁶Mathematical Sciences, University of Southampton, Southampton, UK
⁷Southampton Clinical Trials Unit, University of Southampton, Southampton, UK
⁸Macmillan Survivorship Research Group, University of Southampton, Southampton, UK
⁹Saw Swee Hock Public School of Health, National University of Singapore, Singapore
¹⁰Leeds Institute of Health Sciences, University of Leeds, Leeds, UK
¹¹Dorset Cancer Centre, Poole, UK
¹²Southern Health NHS Foundation Trust, Southampton, UK
¹³School of Nursing and Midwifery, Oxford Brookes University, Oxford, UK
¹⁴Faculty of Health & Life Sciences, Coventry University, Coventry, UK
¹⁵School of Health Sciences, Bangor University, Bangor, UK
¹⁶School of Experimental Psychology, University of Bristol, Bristol, UK

**Acknowledgements** The trial team would like to thank Julie Hooper, Megan Liddiard and Karen Middleton for their assistance with the recruitment of GP surgeries. Eat for Health was largely developed by Tara Cheetham-Blake in collaboration with Joanna Slodkowska-Barabasz, also of the University of Southampton. We would also like to thank Kirsten Smith, Mary Steele and Jin Zhang of the University of Southampton for their work on programming Renewed Online. We would like to thank all of our PPIs who helped in discussions surrounding the study procedures and helped us to develop our intervention on the project: Tamsin Burford, Geoff Sharman, Kevin Summers, Roger Bacon and other members of the PPI team who wish to remain anonymous.

**Contributors** AK drafted the manuscript with input from all authors. LY and PL designed the study and secured funding. LY and KB led the overall development of Renewed online. LY, AK, KB, JB, and TeC developed procedures for the study design with help from LW. LY and KB oversaw the development of all of the intervention and had final approval of all content. AK and AWAG developed resources for Healthy Paths. AM, KS and TeC developed resources for Getting Active. TaC-B developed additional content for men on Active Surveillance. All co-authors and PPI reps (RB, GL, GS, KS) also provided input into the development of Renewed. BS wrote the statistical analysis plan. GLY wrote the plan for the health economic analysis. DB, DE, CF, RDN, RO, SR, AR, EW and CW contributed to the study design as part of the management team and contributed to the final version of the manuscript. All authors approved the final manuscript. AK is the guarantor.

**Funding** This project is funded by the National Institute for Health Research (NIHR) [Programme Grants for Applied Research Programme/Reference Number RP-PG-0514-20001]. The views expressed are those of the authors and not necessarily those of the NIHR or the Department of Health and Social Care (See page 39).

**Competing interests** None declared.

**Patient consent for publication** Not required.

**Ethics approval** This trial was approved in March 2017 by Ethics and Research Governance in Southampton, ID 25160 and has been approved by the NHS Research Ethics Committee via the Integrated Research Application System, ID 238636, N/RES Reference 18/NW/0013.

**Provenance and peer review** Not commissioned; externally peer reviewed.

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
