## [Reviewer comments · BMJ Open]

ARTICLE DETAILS

TITLE (PROVISIONAL)	RENEWED: PROTOCOL FOR A RANDOMISED CONTROLLED TRIAL OF A DIGITAL INTERVENTION TO SUPPORT QUALITY OF LIFE IN CANCER SURVIVORS
AUTHORS	Krusche, Adele; Bradbury, Katherine; Corbett, Teresa; Barnett, Jane; Stuart, Beth; Yao, Guiqing; Bacon, Roger; Böhning, Dankmar; Cheetham-Blake, Tara; Eccles, Diana; Foster, Claire; Geraghty, Adam; Leydon, Geraldine; Muller, Andre; Neal, Richard; Osborne, Richard; Rathod, Shanaya; Richardson, Alison; Sharman, Geoffrey; Summers, Kevin; Watson, Eila; Wilde, Laura; Wilkinson, Clare; Yardley, Lucy; Little, Paul

VERSION 1 – REVIEW

REVIEWER	Reviewer name: Christina Signorelli Institution and Country: Kids Cancer Centre, Sydney Children's Hospital/UNSW Sydney, Australia. Competing interests: None declared.
REVIEW RETURNED	17-Sep-2018

GENERAL COMMENTS	The authors present a study protocol for a randomised controlled trial evaluating the efficacy of a digital intervention to improve cancer survivors' quality of life. Overall the protocol is well written and address a significant gap in research and clinical practice, in terms of minimising the long-term burden of cancer. The research questions are clear and the methods and research design are appropriate. There are a few minor areas which are ambiguous or lack detail, which I have suggested changes to improve the reporting of the protocol in terms of clarity or to facilitate replication of the study in future. Some reordering of the protocol is also needed to improve its readability and flow, which I have proposed throughout. ABSTRACT Overall the abstract offers a succinct summary for the protocol, including a clear rationale and sufficient detail about the methodology. Some very minor suggestions: Page 2, line 7: "Issues relating to low quality of life..." is vague as a standalone observation. Suggest adding brief examples of these issues, or just rewording to "Low quality of life is common in cancer survivors" or something similar. Page 2, line 33: suggest mentioning the name of the QoL measure used if space allows. INTRODUCTION The introduction provides a strong foundation for the study to be carried out, including detailed description of the relevant literature, the research questions and the gap which this research will attempt to fill. Some of the following changes may enhance this section:
---

Generally, there isn't any information about the prevalence of poor QoL in survivors and its impact, potentially including their lasting impact and/or latency in some cases. I think these details would strengthen the case for the 'Renewed' intervention. Understanding the prevalence of secondary cancers would also establish a clearer basis for the fourth research aim for example. Personally, I also would reorder to 1) physical activity, 2) diet, 3) weight loss, and 4) psychological wellbeing (throughout the protocol).

Page 5, lines 42-48: "Two interventions" is possibly a little misleading, since it's the same intervention in essence ('Renewed') and one arm also receives additional human support.

Page 6, line 5: Please define EORTC as this is its first use; possibly check other abbreviations too.

Page 6, lines 35-41: It is unclear how the two qualitative research questions differ. Perhaps elaborate on how patients HCPs "view the Renewed intervention" (RQ1) to make it clearer how this differs from acceptability (RQ2).

METHODS

The methods are suitable for the research questions and generally well detailed, with some minor areas requiring clarity or improvement. The authors should be commended for including the sub-study examining questionnaire design strategies to assess completion rates.

General comments: Some information is duplicated throughout the methods and could be reduced by simply reordering. Perhaps detail the eligibility and recruitment first, describe the intervention itself, followed by data collection and measures, and conclude with the randomisation, sample size/power calculations, and statistical analysis. Also, how will practices be chosen to approach for potential recruitment?

Page 7, lines 16-39: Are consumers still involved? The tense seems to change throughout this paragraph, so it is unclear if they were only consulted during the study design or if they are/will be consulted throughout the conduct of the study throughout.

Page 7, line 44: "The trial will recruit approximately 835 patients per cancer type (total n = 2500)." Suggest moving to sample size/power calculation section, it is not mentioned there. Also, has recruitment commenced and is it still ongoing. Please specify the (expected) dates of recruitment?

Page 8, line 9: Will any other clinical information be collected? Disease stage, risk, treatment types etc. will likely be important in evaluating QoL but also some of the secondary outcomes including economic considerations.

Page 9, line 16: Write out SMD in full, particularly as this is the only use of it.

Page 9, line 49: "by EORTC score (high/low QoL; 64 or less/65 or more taking 65 as the lower 25% CI from previous study data [28])." Should high scores read "65-85" since those with scores >85 will be excluded?

Page 10, line 31: "See Error! Reference source not found for a detailed diagram" I assume this was supposed to link to the Figure, but there seems to be an error. Please amend.

Page 10, line 33-37: The format of the website and participants' involvement is a little vague. Is it designed for single access only or use over time? How do participants "move through" the 'Renewed' content, including the four interventions?

	Are they able to view/participate in any order, or in only some but not all four interventions? At what point is the “information most relevant to specific cancer types” and “links to external resources” presented? Will the varying degrees of involvement between participants be accounted for in terms of the effect on QoL and other outcomes of interest? Further information would be helpful, particularly as some of these interventions have previously been evaluated independently. Page 11, lines 7-20: How will potentially conflicting information between the four interventions be managed? For example, participants may receive advice on a low carb or calorie diet for weight loss via ‘Power+’ and then potentially different advice on healthy eating via ‘Eat for Health’. Or would participants only have access to one intervention over the other depending on their needs? Page 12, lines 29-44: Given the potential for highly varying types/levels of support from the “Supporters”, will these be measured/ensured in any way for consistency? Also, please clarify if the role of the Supporter is to encourage participation generally or assist with the delivery of specific interventions in some way. Page 12, lines 53-55: Will participants who choose not to receive support be analysed in the ‘Renewed only’ or ‘Renewed with support’ group? Page 13, line 9-15: “Other than this, participants in the usual care group will continue to use their existing medical support as they usually would.” Such as? What do we know about the ‘usual care’ patients would receive without an intervention like ‘Renewed’, if any? Page 13, line 43-44: “At study completion, the usual care (waitlist) group will be given access to the Renewed website.” What if the results show that the intervention is only effective with support? Page 14, lines 7-16: Suggest moving information about sub-study analysis to “statistical analysis” section for consistency. Page 14, Table 2: Suggest reordering the footnote explaining acronyms to align with the order of the items as they appear in the table. Page 16, lines 14-22: Please move this information to “Data collection” section for consistency. Page 17, lines 3-48: Regarding the qualitative component of the study, who will conduct the interviews, coding and analysis? What proportion of data will be double coded? Page 18, lines 2-27: Greater consideration is needed regarding the nature of any possible or expected adverse events including potential risks at recruitment, screening or throughout the intervention and during follow-up, even if unlikely. What measures will be put in place to ensure these are managed efficiently and appropriately e.g. support or referrals? FIGURES The figures are not referred to in the text.
--	--

REVIEWER	Reviewer name: Wonshik Chee Institution and Country: Duke University, USA Competing interests: None declared
REVIEW RETURNED	23-Sep-2018

GENERAL COMMENTS	Well written manuscript with comprehensive contents related to issues of digital intervention.
--

VERSION 1 – AUTHOR RESPONSE

Reviewer: 1

Reviewer Name: Christina Signorelli

Institution and Country: Kids Cancer Centre, Sydney Children's Hospital/UNSW Sydney, Australia.

Please state any competing interests or state 'None declared': None declared.

The authors present a study protocol for a randomised controlled trial evaluating the efficacy of a digital intervention to improve cancer survivors' quality of life. Overall the protocol is well written and address a significant gap in research and clinical practice, in terms of minimising the long-term burden of cancer. The research questions are clear and the methods and research design are appropriate. There are a few minor areas which are ambiguous or lack detail, which I have suggested changes to improve the reporting of the protocol in terms of clarity or to facilitate replication of the study in future. Some reordering of the protocol is also needed to improve its readability and flow, which I have proposed throughout.

ABSTRACT

Overall the abstract offers a succinct summary for the protocol, including a clear rationale and sufficient detail about the methodology. Some very minor suggestions:

Page 2, line 7: "Issues relating to low quality of life..." is vague as a standalone observation. Suggest adding brief examples of these issues, or just rewording to "Low quality of life is common in cancer survivors" or something similar.

Page 2, line 33: suggest mentioning the name of the QoL measure used if space allows.

Thank you for these suggestions. We have amended the Abstract accordingly.

INTRODUCTION

The introduction provides a strong foundation for the study to be carried out, including detailed description of the relevant literature, the research questions and the gap which this research will attempt to fill. Some of the following changes may enhance this section:

Generally, there isn't any information about the prevalence of poor QoL in survivors and its impact, potentially including their lasting impact and/or latency in some cases. I think these details would strengthen the case for the 'Renewed' intervention. Understanding the prevalence of secondary cancers would also establish a clearer basis for the fourth research aim for example.

Thank you for this feedback. We have added much more information about the prevalence of QoL issues and more details about the impact as well as secondary cancer prevalence in the Introduction, page 4.

Personally, I also would reorder to 1) physical activity, 2) diet, 3) weight loss, and 4) psychological wellbeing (throughout the protocol).

Thank you for your suggestion. However, we discussed the order in which the different intervention parts would be best presented and this was the consensus.

Page 5, lines 42-48: "Two interventions" is possibly a little misleading, since it's the same intervention in essence ('Renewed') and one arm also receives additional human support.

Thank you, we have amended that sentence to: “The aim of the Renewed trial is to evaluate the effectiveness and cost-effectiveness of two trial arms using an intervention designed to improve QoL in breast, colon and prostate cancer survivors: the first arm will be given access to a digital intervention (Renewed) and the second will be given access to Renewed accompanied by brief human support.”

Page 6, line 5: Please define EORTC as this is its first use; possibly check other abbreviations too.

Page 6, lines 35-41: It is unclear how the two qualitative research questions differ. Perhaps elaborate on how patients HCPs “view the Renewed intervention” (RQ1) to make it clearer how this differs from acceptability (RQ2).

Thank you. We have amended this. The hypotheses now read: “1. Is the Renewed intervention with or without human support, acceptable to patients and healthcare practitioners?

2. Is the Renewed support feasible for healthcare practitioners to implement in practice?” on page 6.

METHODS

The methods are suitable for the research questions and generally well detailed, with some minor areas requiring clarity or improvement. The authors should be commended for including the sub-study examining questionnaire design strategies to assess completion rates.

General comments: Some information is duplicated throughout the methods and could be reduced by simply reordering. Perhaps detail the eligibility and recruitment first, describe the intervention itself, followed by data collection and measures, and conclude with the randomisation, sample size/power calculations, and statistical analysis. Also, how will practices be chosen to approach for potential recruitment?

Thank you for this suggestion. We have amended the information regarding GP surgery participation on page 8. They are being recruited via local Clinical Research Networks.

Regarding the re-ordering of the Methods however, we have presented these in the order of other papers published in this journal to be in keeping with the style. However, if the editor would prefer us to change this then we would be happy to oblige.

Page 7, lines 16-39: Are consumers still involved? The tense seems to change throughout this paragraph, so it is unclear if they were only consulted during the study design or if they are/will be consulted throughout the conduct of the+ study throughout.

We have amended this thoroughly and have provided much more detail about the PPIs involved in the study on page 7.

Page 7, line 44: “The trial will recruit approximately 835 patients per cancer type (total n = 2500).” Suggest moving to sample size/power calculation section, it is not mentioned there. Also, has recruitment commenced and is it still ongoing. Please specify the (expected) dates of recruitment?

We have moved this information to the paragraph on power calculation (pages 9-10) as suggested and added our expected recruitment dates to this section.

Page 8, line 9: Will any other clinical information be collected? Disease stage, risk, treatment types etc. will likely be important in evaluating QoL but also some of the secondary outcomes including economic considerations.

Although we have listed our exclusion only here, we will also be performing a notes review for each participant at the end of the study to collect other clinical data such that you describe; page 17 outlines this and the Health Economics analysis in some detail.

Page 9, line 16: Write out SMD in full, particularly as this is the only use of it.

Thank you for pointing this out. We have amended this.

Page 9, line 49: "by EORTC score (high/low QoL; 64 or less/65 or more taking 65 as the lower 25% CI from previous study data [28])." Should high scores read "65-85" since those with scores >85 will be excluded?

We have updated this paragraph to reflect this, thank you, now on page 10.

Page 10, line 31: "See Error! Reference source not found for a detailed diagram" I assume this was supposed to link to the Figure, but there seems to be an error. Please amend.

Apologies for this. We have updated it.

Page 10, line 33-37: The format of the website and participants' involvement is a little vague. Is it designed for single access only or use over time? How do participants "move through" the 'Renewed' content, including the four interventions? Are they able to view/participate in any order, or in only some but not all four interventions? At what point is the "information most relevant to specific cancer types" and "links to external resources" presented? Will the varying degrees of involvement between participants be accounted for in terms of the effect on QoL and other outcomes of interest? Further information would be helpful, particularly as some of these interventions have previously been evaluated independently.

Thank you for this feedback. We have added a lot of detail to explain more about what the Renewed intervention entails, on page 11. Participants move through the introductory pages and are then able to choose which parts they would like to look at in more detail. Further, participants are able to log off and log in again as they desire.

Tailoring is included throughout and we have provided further examples. While we have included tailoring in this intervention, we also wanted to create an intervention that is potentially generalizable to various cancer types and have only included tailoring by cancer type where cancer-specific lifestyle advice would directly impact behaviour change. Links to other resources are also outlined in more detail, for instance, we provide links to informational and supportive websites about returning to work on our homepage.

The varying degrees of participation in the intervention will be accounted for during the process analysis, when we will gather and analyse data on how people use the website, e.g. which pages have been viewed and how this relates to outcomes.

Page 11, lines 7-20: How will potentially conflicting information between the four interventions be managed? For example, participants may receive advice on a low carb or calorie diet for weight loss via 'Power+' and then potentially different advice on healthy eating via 'Eat for Health'. Or would participants only have access to one intervention over the other depending on their needs?

We have clarified this on page 11:- From the 96 interviews with patients who used our prototype intervention we think it is unlikely that people would choose to use both POWeR+ and Eat for Health, but have designed these in such a way that they do not contain conflicting dietary information. For example the low carb diet in POWeR encourages eating less red/processed meat, despite these meats being low in carbohydrates, because of the link between these kinds of meat and cancer. Therefore if someone used POWeR and Eat for health they would find a similar overall diet.

However, POWeR has a lot of focus on self-regulatory skills which can help people to lose weight (e.g. how to set up your environment to lose weight). Eat for Health includes recipe ideas and guidance as to what foods are healthier and how to eat healthily, taking concerns such as money, into account, whereas POWeR+ specifically has guidance on how to lose weight.

Page 12, lines 29-44: Given the potential for highly varying types/levels of support from the “Supporters”, will these be measured/ensured in any way for consistency? Also, please clarify if the role of the Supporter is to encourage participation generally or assist with the delivery of specific interventions in some way.

We have clarified this on (page 13-14): Supporters will log all support given on a document provided by the research team. These logs are collected and the data will be analysed.

We have added more information to page 13. Supporters are asked to encourage participants to use the intervention as part of the CARE (congratulate, ask, reassure, encourage) approach, as we state there this could be encouragement to continue to use the website or try out/continue an offline behaviour change (e.g. increasing physical activity). The Supporter does not provide or assist with any other part of the intervention beyond what is described, they just deliver the CARE approach to support autonomous motivation.

Page 12, lines 53-55: Will participants who choose not to receive support be analysed in the ‘Renewed only’ or ‘Renewed with support’ group?

Given that all participants in the group who receive support will receive three emails from Supporters, there is a difference between the two groups, even if patients choose not to ask for additional telephone support. When analysing the data, the amount and type of support received will be taken into account.

Page 13, line 9-15: “Other than this, participants in the usual care group will continue to use their existing medical support as they usually would.” Such as? What do we know about the ‘usual care’ patients would receive without an intervention like ‘Renewed’, if any?

Usual care is a term used to describe any care that participants would usually access, we do not have details on what this is likely to be, although there is no standard ‘usual care’ for cancer survivors in the UK, so it may well be nothing for most participants, we plan to ask participants what care they received whilst in the study to find out more about this. We have added a sentence: “At the end of the study, participants will be asked what other interventions, if any, they have used while taking part in the study.” to page 14.

Page 13, line 43-44: “At study completion, the usual care (waitlist) group will be given access to the Renewed website.” What if the results show that the intervention is only effective with support?

We will not know, at this stage, whether support is important for change. Participants will be recruited over at least a year so usual care participants will be offered the intervention when they complete the final questionnaires, which will be some time before we have completed collecting all data and analysed it. If support is important when the analysis has been conducted, obviously this will have implications for the treatment of cancer survivors and dissemination. It is beyond the scope of this study to offer support to usual care participants at study end but we are hopeful that people will find some advice useful should they choose to use Renewed at that point. We will not be analysing the results of usual care patients using Renewed at study end as part of this project.

Page 14, lines 7-16: Suggest moving information about sub-study analysis to “statistical analysis” section for consistency.

As per your advice, we have moved this information to the Statistical Analysis section on pages 16-17.

Page 14, Table 2: Suggest reordering the footnote explaining acronyms to align with the order of the items as they appear in the table.

Thank you for pointing this out. We have changed the order accordingly.

Page 16, lines 14-22: Please move this information to “Data collection” section for consistency.

We have moved this to Data Collection on page 14, as suggested.

Page 17, lines 3-48: Regarding the qualitative component of the study, who will conduct the interviews, coding and analysis? What proportion of data will be double coded?

The interviews and analysis will be conducted by members of the research team; this will consist of a PhD student, senior research assistants and post-doctoral research fellows, we cannot name them at this point. We have not proposed to double code the data so have not provided detail on this.

Page 18, lines 2-27: Greater consideration is needed regarding the nature of any possible or expected adverse events including potential risks at recruitment, screening or throughout the intervention and during follow-up, even if unlikely. What measures will be put in place to ensure these are managed efficiently and appropriately e.g. support or referrals?

There should be no adverse events at recruitment as the GP surgeries conduct the searches and screen for anyone who may be unlikely to be eligible (e.g. lack of capacity or a more severe psychological issue).

We have added the following information to page 19: “It is very unlikely that there will be any adverse events during screening and questionnaire completion, given that screening online consists of answering questions, however, participants are repeatedly told that they can contact the research team who can signpost to other support services as appropriate.” and “Should a participant be ineligible for the study for any reason, they are presented with a page of alternative support services.”

If, at any time, any member of the research team is made aware of any adverse event, depending on the nature of said event, the research team will discuss it and decide on which form of action to take.

FIGURES

The figures are not referred to in the text.

Apologies. This was the error. We have updated this now and referred to the other figure in the text.

Reviewer: 2

Reviewer Name: Wonshik Chee

Institution and Country: Duke University, USA

Please state any competing interests or state ‘None declared’: None declared

Well written manuscript with comprehensive contents related to issues of digital intervention.

Your feedback is very much appreciated. Thank you.

VERSION 2 – REVIEW

REVIEWER	Reviewer name: Christina Signorelli Institution and Country: Kids Cancer Centre, Sydney Children's Hospital/UNSW Sydney, Australia Competing interests: None declared
REVIEW RETURNED	16-Nov-2018

GENERAL COMMENTS	The authors have adequately addressed the recommendations and made revisions to the manuscript where appropriate. I have no further comments or suggested changes.
--